# I-LORA: ITERATIVE MERGING OF ROUTING-TUNED LOW-RANK ADAPTERS FOR MULTI-TASK LEARNING

## ABSTRACT

The advancement of vision-language models has significantly boosted the performance of embodied and game AI, endowing them with stronger general visual understanding capabilities and logical abilities for action planning. However, the substantial computational cost of model training and the performance degradation during fine-tuning limit the models' ability to learn emerging new tasks continually. Creating a versatile and dynamically updatable vision-language model is an essential area of research. To this end, we propose a Low-Rank Adapter-based fine-tuning approach called I-LoRA, which enables iterative and independent learning of new tasks while preserving the logical capabilities of the previously trained model. Specifically, we first design the routing-tuning method to minimize the impact of original capabilities from the new task by minimizing activation values of LoRA matrices as low as possible in the general task. Secondly, we propose a novel approach to iteratively merge new adapters, allowing for continuous integration of adapters trained on new tasks without being influenced by task order, thereby reducing interference between them. Finally, we conducted extensive experiments on public datasets with significant behavioral and logical differences between tasks. The results demonstrate that our approach achieves excellent single-task performance, strong multi-task compatibility, and flexible scalability without increasing the number of model parameters.

## 1 INTRODUCTION

The powerful logical reasoning and general visual understanding capabilities of vision-language models (VLMs) have yielded significant progress in embodied and game AI (Wang et al., 2023a; Driess et al., 2023). Beyond their ability to perform in-context learning directly, pre-trained models possess extensive general knowledge that can be effectively leveraged for tuning on domain-specific tasks. Vision-language-action (VLA) models, derived from pre-trained vision-language models, can be further trained to learn and adapt to the behavioral logic of agents Kim et al. (2024); Ma et al. (2024). Fine-tuning these pre-trained models showcases the potential of VLMs to acquire new behavioral logic. However, in practical applications, the range of tasks continues to grow, and the substantial computational cost of model training, coupled with performance degradation during fine-tuning, poses significant challenges to their ability to continually learn new tasks. Thus, creating an approach to versatilely and dynamically update the vision-language model to efficiently adapt new tasks is important.

In the literature, the focus of building general embodied and game AI using large language models has been on constructing general agents. For example, some work attempts to build a lifelong learning agent power by LLM, (Wang et al., 2023a) proposed an iterative prompting mechanism and an increasing database for LLM to retrieve. In contrast, (Fan et al., 2022) utilized an Internet-scale knowledge base, with both conducting experiments in Minecraft. Their commonality lies in expanding the model's knowledge base, using LLM's reasoning capabilities to execute action planning. However, this approach is limited in adapting to new task logic and requires high costs in inferencing and decision-making. While fine-tuning allows the model to adapt to different tasks based on new data and can significantly improve decision efficiency, it often comes at the expense of the model's original task capabilities.

Mastering a particular skill often requires fine-tuning the model. LoRA (Low-Rank Adaptation) adapters, used by merging into the base model when inference, can naturally become modular for LLM, which is inherently convenient for model fusion. So, we used LoRA matrices to fine-tune the model for each task. Current approaches to combine LoRA adapters usually involve setting up a router to perform weighted sum over the adapters, (Dou et al., 2023) tries to combine mixture-of-experts(MOE) with LoRA adapters fusion. Still, this method introduces additional parameters and requires further tuning to create new routing mechanisms for each new task adapter, which is nearly impossible for iterative fusion. Additionally, some works have proposed methods for directly combining models without additional training. (Yu et al., 2024) performs random drop and rescale of learned parameters to combine different learned parameters. However, none of these approaches consider how to mitigate the interference between model parameters across different tasks. This issue becomes particularly pronounced when the gap between functions is significant, as directly merging the trained parameter matrices does not yield satisfactory results.

To this end, this paper focuses on enabling the model to learn new tasks while maintaining performance on other tasks. Specifically, we aim to minimize the impact of learning new tasks on the base model parameters, thereby ensuring the composability of learned parameters. To achieve this, we propose a method: Routing Tuning. We redesign the loss function during fine-tuning by injecting some general-purpose data into the training data. When routing from general data, we encourage the model's activation values on this general data to approach 0 using KL-divergence loss and L2-norm of LoRA layer output. This ensures minimal interference with the model's performance on other tasks. Thus, the LoRA adopter can be merged into the base model and retain the routing ability without increasing the parameters.

Subsequently, we propose a novel iterative fusion method to reduce parameter interference during fusion. By employing Singular Value Decomposition (SVD), we further reduce the number of redundant parameters and then select the components with the most significant influence on parameter interference to ensure task performance. We list the properties needed to achieve iterative fusion for LoRA adapters, and then, followed by the properties, we choose the Maximize function to merge parameters in different models. Our merging approach ensures the equality, associativity, and saliency of different tasks, thereby guaranteeing the effectiveness and iterability of task integration. Finally, we conducted extensive experiments in the Atari environment, which included many tasks with significant logical differences. We performed both single-task performance experiments and multi-task fused experiments. The experimental results clearly validate the effectiveness of I-Lora in single-task training as well as its capability for multi-task integration compared with various baselines.

In summary, the contributions of this article can be summarized as follows:

- We have developed a new fine-tuning method called Routing Tuning. This method involves directing the task data and general data to different losses during fine-tuning. As a result, the model is able to learn the specific task while still maintaining its general performance. This approach helps to reduce the interference caused by varying model parameters.

- We have developed a new method to fuse different LoRA adapters into a single entity using SVD and a maximization function, merging them into the base model to maintain performance on specific tasks while preserving general performance. Our approach introduces no additional parameters for the iterative merging of LoRA adapters.

- We collected Atari data for vision-language model training. Using vision-language models, we perform LoRA routing-tuning on a vision-language model for each game, followed by LoRA adapters' composition to validate the iterative merging methods' effectiveness.

## 2 RELATED WORK

Our research focuses on multimodal large language models for gaming and embodied AI, as well as multitask learning and knowledge fusion using low-rank adaptation methods. The most pertinent studies are outlined below.

## 2.1 Vision-language Models on Game and Embodied AI

LLMs have excellent learning and reasoning abilities (Achiam et al., 2023), and vision-language models(VLMs) have shown remarkable capabilities in multiple multimodal tasks such as visual question answering and image captioning (Liu et al., 2024a). Some works have explored the application of vision-language models in games (Wang et al., 2023b). In the literature, VLMs are extended to act as agents to plan complex actions for intricate tasks(, FAIR). (Hu et al., 2024) builds an LLM-embodied agent accessing external knowledge for in-context learning to play Pokemon battles. VLMs for games input images and text-like game rules to obtain the corresponding plan and action. Unlike games, embodied AI must control physical entities and interact with the environment. Vision-language-action models(VLAs) handle multi-modal inputs and generate actions for robots to complete embodied tasks. VIMA (Jiang et al., 2022) introduces different multi-modal prompts compared to traditional pure text prompts to generalize the model to multiple tasks. RT-2 (Brohan et al., 2023) utilizes large multimodal models for robotics tasks, introduces the co-fine-tuning technique, and reserves the model's general ability while learning robotics data.

Building a model that handles multiple embodied tasks and continues to explore and develop new skills is crucial for achieving general AI models. (Reed et al., 2022) propose unifying the form of inputs and outputs of multiple tasks by designing tokenization and embedding methods. Then, they use autoregressive training to build a general agent. But it's still hard to continue learning because training on new tasks will cause catastrophic forgetting of previous tasks. Building agents with an expandable knowledge base is feasible because large language models have robust retrieval and reasoning abilities. VOYAGER(Wang et al., 2023a) proposes a lifelong learning agent for Minecraft with GPT models. They utilize LLMs to discover diverse tasks, update the task libraries, and prompt using in-context learning. However, they all depend on human-labeled data, do not introduce any updates to the capabilities of the LLMs.

Inspired by previous work, we employ vision-language models to address multiple game tasks, which avoids the need for designing complex rules and expert knowledge required by agent-based methods. By leveraging the model's inherent knowledge and optimization capabilities and based on the knowledge fusion capability of LLM, we can build a continuous learning model for multiple tasks.

## 2.2 Model Merging and Lora Adapters Fusion

In the literature, there have been many attempts at model merging in academia, (Yang et al., 2024) provides a detailed survey of the current model merging methods. (Yang et al., 2023) learns a distinct merging coefficient layer-wise or task-wise to merge task parameters. (Zhang et al., 2024) and (Ilharco et al., 2022) address knowledge composition and model editing through task vectors, which can also be applied to model merging.

To adapt LLM to different tasks, continual pre-training, and fine-tuning are essential. VLMs are extremely large, and fully finetuning is quite computationally expensive. In addition to that, using parameter-efficient methods like LoRA(Hu et al., 2021) can make fine-tuning more feasible. Different fine-tuned adapters can be combined with different weights to solve intricate tasks. Some research attempts to combine LoRA with a mixture of experts(MoE)(Dou et al., 2023) to generate a combination of different weights for different tasks. Most of the research(Huang et al., 2023) is focused on improving weight combination ways, which introduces extra parameters and computational consumption. Some research also focuses on combining different task matrices into a single matrix. DARE(Yu et al., 2024) randomly drops fine-tuned parameters and rescales the remaining ones, then a different matrix is added together. Merging methods like DARE are implemented in the PEFT(Mangrulkar et al., 2022) (Zhang et al., 2023)library.

Based on previous research, we propose a new way to merge LoRA weights. The method needs to introduce no extra parameters and reserve general capabilities. Also we need the model to iterative merge new parameters which can be further expand to new tasks.

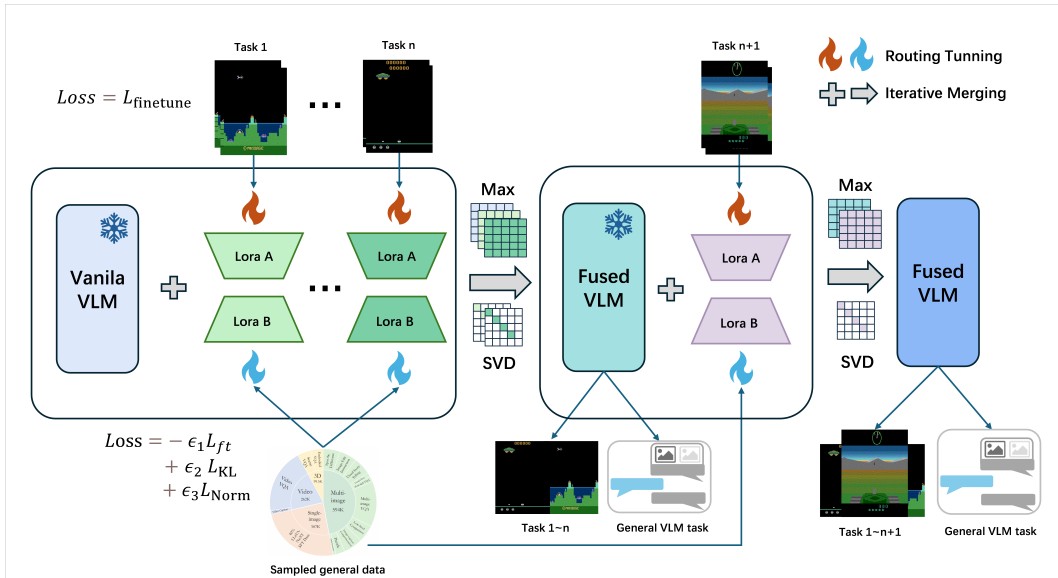

Figure 1: Overview framework of I-LoRA. For each game, we use routing tuning to get a LoRA adapters as shown in the left part. Then we merge a new LoRA adapter to the previous fused adapters to achieve iterative fusion.

## 3 METHODS

### 3.1 PROBLEM FORMULATION AND NOTATION

Given a set of tasks $\tau = \{t_1, \dots, t_T\}$, each task has a set of examples $\mathcal{D}_i = \{(\mathbf{x}_1, \mathbf{y}_1), \dots, (\mathbf{x}_n, \mathbf{y}_n)\}$. Multitask data are the union of the training set of given tasks. We also sample a general dataset $\mathcal{D}^{\mathcal{G}}$ that contains the general training set of vision language models like LLaVA (Liu et al., 2024a) and general game data sampled from publicly available datasets from the Internet.

For each task, we train a LoRA adapter $\{A_i, B_i\}$ to ensure good performance on the corresponding task when inference as follows:

$$h = W\mathbf{x} + s \cdot A_i B_i^\top \mathbf{x}. \tag{1}$$

Subsequently, we incorporate general purpose data to obtain a composable LoRA adapter $\{A_i^{\mathcal{G}}, B_i^{\mathcal{G}}\}$, enabling it to perform well in the specific task without degrading general performance. Finally, we attempt to combine adapters from multiple tasks, aiming to maintain the performance of individual tasks without significant degradation while retaining strong generalization capabilities.

### 3.2 OVERVIEW

Our approach aims to develop a general-purpose model that can handle multiple tasks and iteratively update. To achieve this goal, we often encounter fine-tuned models and a set of task-specific training data. Our methods consist of two parts:

- **Routing Tunning**: This method aims to enhance the generalization capability of the fine-tuned model, ensuring that the fine-tuned LoRA adapter is only adequate for the current task. At the same time, its activation is constrained to zero when dealing with general or other tasks. This is achieved by routing different data to different losses and learning separately.

- **Iterative Merging**: We design a method to integrate the newly fine-tuned adapter, derived from the additional training data, with previously integrated adapters. This integration aims to maintain performance across multiple tasks without compromising the model's general capabilities.

The following introduction to our methodology will be divided into two parts. First, we will explain our Routing tuning method to minimize the impact of fine-tuning on the model's generalization abilities. Next, we will describe how to efficiently and iteratively integrate multiple task-specific fine-tuned LoRA adapters, thereby preserving performance on individual tasks while maintaining the model's general capabilities.

### 3.3 RESERVE GENERAL CAPABILITIES WITH ROUTING TUNNING

For the given task and the data set $\mathcal{D}_i = \{(\mathbf{x}_1, \mathbf{y}_1), \ldots, (\mathbf{x}_n, \mathbf{y}_n)\}$, we fine-tuned the model using LoRA, getting a set of adapters $\mathcal{A} = \{(A_1, B_1), \cdots, (A_n, B_n)\}$ achieving improved performance. However, we observed a significant degradation in the generalizability of the model. Due to the additive property of LoRA matrices, we attempted to merge the LoRA adapters from multiple tasks by summing them. We want the sum of adapters that achieve $(A_1 B_1 + \cdots + A_n B_n) \circ X_i \simeq A_i B_i \circ X_i$.

We experimented with LoRA fusion methods provided by the PEFT (Mangrulkar et al., 2022) library, including DARE (Yu et al., 2024) and TIES (Yadav et al., 2024). In all cases, we observed a considerable decrease in performance, suggesting interference between parameters from different tasks. This led us to consider strategies to mitigate such interference.

Inspired by model-unlearning(Yao et al., 2023; Liu et al., 2024b) techniques, we naturally considered incorporating some general datasets into the training set. These datasets encompass multiple tasks, and by constraining the activation of the LoRA matrices on these tasks, we can alleviate the degradation in generalization performance while mitigating the interference between parameters across different tasks. Consequently, we designed two loss functions.

During training, we first identify the data in the batch that does not belong to the target task and exclude it from the loss calculation of the language modeling. Then we design $\mathcal{L}_{\text{KL}}$ and $\mathcal{L}_{\text{norm}}$. The $\mathcal{L}_{\text{KL}}$ computes the KL divergence between the outputs of the base model and the model augmented with LoRA to ensure that they remain as consistent as possible, preventing the model updates from affecting their original outputs. Furthermore, $\mathcal{L}_{\text{norm}}$ captures the output of the LoRA_B layer during inference on general datasets, ensuring that LoRA activations in general tasks approach zero, thus preserving the performance of the original model. The parameter updating process is summarized by:

$$\theta_{t+1} \leftarrow \theta_t - \underbrace{\epsilon_1 \cdot \nabla_{\theta_t} \mathcal{L}_{\text{lm}}}_{\text{Finetune Loss}} - \underbrace{\epsilon_2 \cdot \nabla_{\theta_t} \mathcal{L}_{\text{KL}}}_{\text{KL loss}} - \underbrace{\epsilon_3 \cdot \nabla_{\theta_t} \mathcal{L}_{\text{norm}}}_{\text{L2 norm of LoRA output}} , \tag{2}$$

$$\mathcal{L}_{\text{KL}} := \sum_{(x^{\mathcal{G}}, y^{\mathcal{G}}) \in D^{\mathcal{G}}} \sum_{i=1}^{|y^{\mathcal{G}}|} \mathcal{KL}\left(h_{\theta^{\circ}}\left(x^{\mathcal{G}}, y^{\mathcal{G}}_{<i}\right) \| h_{\theta_t}\left(x^{\mathcal{G}}, y^{\mathcal{G}}_{<i}\right)\right), \tag{3}$$

$$\mathcal{L}_{\text{norm}} = \sum_{(x^{\mathcal{G}}, y^{\mathcal{G}}) \in D^{\mathcal{G}}} \sum_{i=1}^{|LoRALayers|} \text{Avg}\left(\|Output_{Lora\_B}\|^2\right), \tag{4}$$

where $\theta_t$ indicates the parameters of LoRA adapters at the t-th step, $\mathcal{L}_{\text{lm}}$ is the language modeling loss during fine-tuning, we use the compute_loss function from huggingface training pipeline, the process is shown in Figure 1.

### 3.4 ITERATIVE MERGING LORA ADAPTERS

After Routing Tunning, we get a variety of LoRA adapters for different tasks. We endeavored to integrate them by minimizing the interference of parameters. We first propose to reduce the redundant LoRA parameters further. Based on (Yadav et al., 2024), we can use SVD to reduce parameter interference. So our initial approach involved conducting Singular Value Decomposition (SVD) on the LoRA parameters to mitigate interference between the different adapters. This process is represented in fig 2. Our experiments revealed that the proportion of singular values retained did

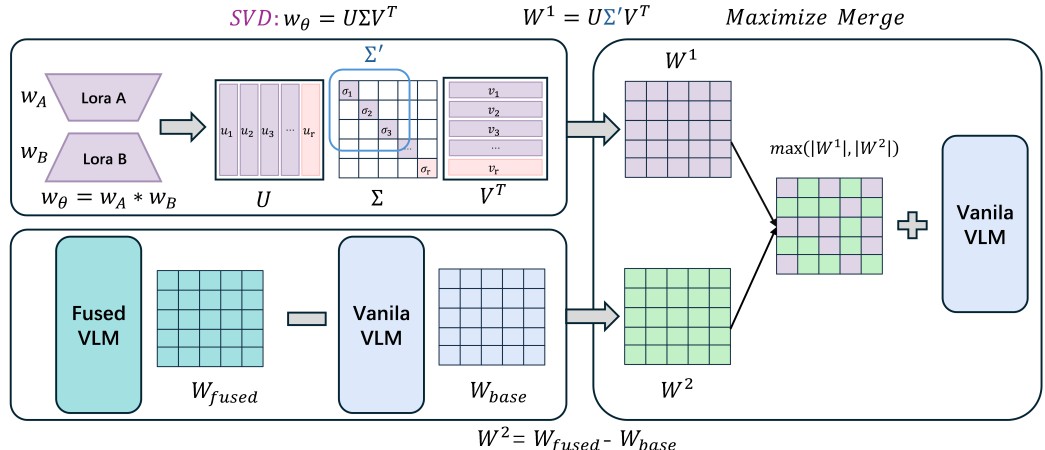

Figure 2: Workflow of iterative merging method.

not significantly impact the performance. We decided to maintain 80% of the singular values based on our findings to construct the new parameter matrix. The process is described as follows:

$$
\begin{aligned}
A &= U\Sigma V^{\top}, \\
\Sigma &= \operatorname{diag}\left(\sigma_1, \sigma_2, \ldots, \sigma_r\right), \\
p &= \lceil 0.8 \times r \rceil, \\
\Sigma' &= \operatorname{diag}\left(\sigma_1, \sigma_2, \ldots, \sigma_p, 0, \ldots, 0\right), \\
A' &= U\Sigma'V^{\top},
\end{aligned}
\tag{5}
$$

Where $A$ is the multiply of LoRA_A and LoRA_B layers' parameters, we decompose it using SVD and discard the last 20% of singular values by setting them to 0.

The first merging of adapter parameters from two single-task adapters will be decomposed using SVD, and the rank corresponding to the 0.8 quantiles of the singular values will be used. For iterative updates, i.e., when merging an adapter that has already integrated multiple adapters with a new adapter from Routing Tuning, to ensure that the previously fused task parameters are not significantly affected, we retain all singular values in the merged adapter that are larger than the 0.8 quantiles of the singular values in the new adapter.

In the previous section, we used Singular Value Decomposition (SVD) to reduce redundancy and minimize interference between different LoRA adapters. Next, we considered how to achieve iterative fusion. We first discuss the desired properties of dynamic fusion:

1. Equality: Each adapter should have an equal opportunity to participate in the fusion process. Every parameter should equally consider the values provided by each adapter.

2. Associativity: The fusion should satisfy $(A + B) + C = A + B + C$. When merging a new adapter with previously fused adapters, the result should be equivalent to merging all adapters for the first time.

3. Saliency: Given an input, the adapter with larger activation values should stand out more prominently, ensuring the quality of the generated output.

Based on these considerations, linear addition satisfies these properties. However, when two sets of parameters need to be merged into the base model, their combined effect will amplify the perturbation on the original parameters if both share the same sign (positive or negative). Conversely, if they have opposite signs, they will mutually weaken each other's perturbation on the original model parameters, leading to significant performance degradation. Suppose the values of the LoRA adapter parameters are too large or too small. In that case, it degrades the model's generation quality on fine-tuned tasks and significantly reduces the model's generalization performance.

We introduce a property that maximizes the distance from zero to address this. Since we have limited the activation value of other tasks through routing fine-tuning on a single task, we choose the parameter that has the most influence on the model parameter, which is the value farthest from 0, to maximize the impact of each fine-tuned model on the base model's parameters, thereby maximizing their effectiveness. After getting multiple parameters that can be added to the base model $\{\mathbf{W}_1, \mathbf{W}_2, \ldots, \mathbf{W}_n\}$, we will get the final $\mathbf{W}$ by:

$$\mathbf{W}_{ij} = \max\left(\left|\mathbf{W}_{ij}^1\right|, \ldots, \left|\mathbf{W}_{ij}^n\right|\right).\tag{6}$$

After getting the final $\mathbf{W}$, we will add them directly to the corresponding part of the base model to get the new model.

## 4 EXPERIMENTS

Our research involves conducting experiments across a variety of Atari games to assess the efficacy of adapter fusion. Initially, we evaluate the performance of individual adapters on single games. Subsequently, we merge several adapters to examine the resultant performance enhancement.

### 4.1 EXPERIMENTS SETTINGS

**Datasets and Tasks**. We decided on the Atari environment as our experimental setting because Atari contains numerous games with diverse visual rules and varying skill requirements, making it well-suited for exploring the effects of multi-task composition while also providing efficiently quantifiable performance metrics. This makes it a suitable benchmark for evaluating multitask integration capabilities. Specifically, we conducted experiments using Atari games from the Gym(Brockman et al., 2016) environment. We found that no open-source dataset retains the original resolution of Atari environment images. Most reinforcement learning (RL) models are trained on downsampled grayscale images, which are even challenging for humans to interpret, bringing challenges to model learning. Therefore, we opted to sample from checkpoints of models trained on two billons frames on the APPO algorithm using the Sample-Factory(Petrenko et al., 2020) framework. We removed extraneous frames at the beginning and end, which led to failures, collecting 100,000 images per game to construct the images for the dataset.We combined the publicly available Atari challenge dataset with the LLava model's training data to create a general-purpose dataset that maintains the model's generalization ability.

We compiled game rules and corresponding actions in the Gym environment as text input for the dataset, organizing them based on the game rules and the meaning of the action. For the model's ground truth output, we directed it to generate a chain-of-thought reasoning process. This involved identifying the objects in the scene and then determining the appropriate action based on the rules. Our experiments demonstrated that generating a chain-of-thought output resulted in better performance than directly outputting actions.

**Model**. We use LLaVA-interleave-qwen-7b (Li et al., 2024) as our base model. To ensure the model can detect moving objects in the game frame, we need to interleave images for analysis. After testing the game dataset with multiple models, we found that llava-interleave performed the best. For LoRA finetuning, we used 128 as lora_rank and 256 as lora_alpha. For easy adapter fusion, we finetune the vision-language model on every linear module, including vision_encoder, vision_projector, and language models. So, all the adapters can be added together and further merge into the base model directly without introducing extra parameters.

**Evaluation Metrics** For single-game performance, we use mean and median of the human-normalized score, which represents how well the model performs compared to human players, is calculated by $\frac{\text{score}_{\text{agent}} - \text{score}_{\text{random}}}{\text{score}_{\text{human}} - \text{score}_{\text{random}}}$. For multiple LoRA fusion, we use the degradation ratio to measure the performance, which is $\frac{\text{score}_{\text{fused}}}{\text{score}_{\text{original}}}$.

Table 1: Performance of each single task.

| Game Name | Random | Human | RL models | | | Ours |
|-----------|--------|-------|-----------|-----------|------|------|
| | | | SPR | DreamerV3 | DART | |
| Alien | 227.8 | 7172.7 | 841.9 | 959 | 962.0 | 586.7 |
| Amidar | 5.8 | 1719.5 | 179.7 | 139 | 125.7 | 57.0 |
| Assault | 222.4 | 742.0 | 565.6 | 706 | 1316.0 | 3580 |
| Asterix | 210.0 | 8503.3 | 962.5 | 932 | 956.2 | 1200 |
| BattleZone | 2360.0 | 37187.5 | 14834.1 | 12250 | 15325.0 | 16750.0 |
| Boxing | 0.1 | 12.1 | 35.7 | 78 | 83.0 | 65.0 |
| ChopperCommand | 811.0 | 7387.8 | 946.3 | 420 | 1263.8 | 5766.6 |
| CrazyClimber | 10780.5 | 35829.4 | 36700.5 | 97190 | 34070.6 | 39000 |
| DemonAttack | 152.1 | 1971.0 | 517.6 | 303 | 2452.3 | 28718.8 |
| Freeway | 0.0 | 29.6 | 19.3 | 0 | 32.2 | 26.0 |
| Frostbite | 65.2 | 4334.7 | 1170.7 | 909 | 346.8 | 2223.3 |
| Gopher | 257.6 | 2412.5 | 660.6 | 3730 | 1980.5 | 320 |
| Hero | 1027.0 | 30826.4 | 5858.6 | 11161 | 4927.0 | 6395 |
| KungFuMaster | 258.5 | 22736.3 | 14783.2 | 21420 | 23744.3 | 22900 |
| MsPacman | 307.3 | 6951.6 | 1318.4 | 1327 | 1132.7 | 1320 |
| Qbert | 163.9 | 13455.0 | 866.3 | 3405 | 750.9 | 750.0 |
| #Superhuman(↑) | 0 | N/A | 2 | 3 | 5 | **5** |
| Mean(↑) | 0.000 | 1.000 | 0.477 | 0.934 | 0.951 | **2.054** |
| Median(↑) | 0.000 | 1.000 | 0.194 | 0.221 | 0.252 | **0.459** |

## 4.2 EXPERIMENTS RESULTS

### 4.2.1 SINGLE MODEL PERFORMANCE

We follow the setting of an existing Atari 100k benchmark(Kaiser et al., 2019), widely used in reinforcement learning algorithms to test sample efficiency. It consists of 26 games, each being allowed to be trained only in 100K steps. We fine-tuned the model for each game using LoRA, setting the LoRA rank to 128 and the LoRA alpha to 256. Each dataset underwent fine-tuning for five epochs.

We used the mean and median of the human-normalized score to compare with human players, with the total reward from the environment representing the score. For benchmarking, we selected three RL algorithms - SPR (Schwarzer et al., 2020), DreamerV3 (Hafner et al., 2023), and DART (Agarwal et al., 2024) - and obtained results after 100K sampling steps. The results are presented in the table 1.

Based on the experimental results, our model achieved twice the performance comparable to state-of-the-art (SOTA) RL models on the mean and medium of the human-normalized score and outperformed them in some games. And 5 out 16 game outperforms human level. Our findings include:

- Data quality is crucial. Our model also shows improved performance for games where the sampled RL models perform well.
- Atari games exhibit randomness, leading to high variance in validation results. We addressed this by averaging over multiple inference runs.
- Vision-language models can learn to understand the rules of the game directly, achieving better results than the sampled RL models.

### 4.2.2 ROUTING TUNNING PERFORMANCE

The preceding section demonstrates our success in constructing a dataset and the efficacy of vision-language models in addressing Atari tasks. Subsequently, we applied our Routing Tuning method

Table 2: Effectiveness of routing tunning on task data. It shows that all score are above the 0.85% of the original score which shows that our routing tunning has few influence on task performance.

| | ChopperCommand | Asterix | BattleZone | CrazyClimber | Frostbite |
|---|---|---|---|---|---|
| Single Task Finetune | 5766 | 1200 | 16750 | 39000 | 2223.3 |
| Routing Tunning | 5560 | 1370 | 14500 | 44066.7 | 1943 |
| Reserve Rate | 0.96 | 1.14 | 0.86 | 1.12 | 0.87 |

Table 3: Effectiveness of routing tuning on benchmark. SD: Spot the Difference, IE: Image Edit Instruction, QB: Q-Bench, Math: MathVerse-mv. The benckmark of routing tunning and directly fine-tuning shows that routing tunning retain more general capabilities.

| Model | SD | IE | NLVR2 | BW | QB | NLVR2_M | HQ-Edit | MagicBrush | Mantis | Math |
|---|---|---|---|---|---|---|---|---|---|---|
| Base Model | 0.3736 | 0.3117 | 0.8787 | 0.3736 | 0.6923 | 0.8787 | 0.2875 | 0.3359 | 0.5852 | 0.3702 |
| Finetune Single Game | 0.3582 | 0.2939 | 0.8626 | 0.3582 | 0.6923 | 0.8626 | 0.2730 | 0.3148 | 0.5668 | 0.2259 |
| Routing Tunning | 0.3697 | 0.2974 | 0.8557 | 0.3697 | 0.7692 | 0.8557 | 0.2755 | 0.3192 | 0.5714 | 0.3028 |

to fine-tune multiple games. We collected general data from the LLaVA training data and extracted game data from the Atari Grand Challenge dataset(Kurin et al., 2017). The final dataset comprises 100,000 task data, 30,000 general data, and 30,000 other game data.

After Routing Tunning, we test them on task performance and general performance. We compare the task performance with a model that finetuned on task data only by calculating the proportion of their performance. The results are shown in the table 2. Our Routing Tuning method achieved an average score of 99% compared to individually trained models across multiple games, demonstrating that our Routing Tuning does not negatively affect the learning of task-specific data.

To test the general performance, we first run our experiments on the LLaVA-Interleave-Benchmark (Li et al., 2024) provided by the model; then, we compare them on the routing finetuned models. The results are shown in Table 3. Compared to the base model, both single-task fine-tuning and Routing Tuning reduce the model's performance on general datasets. However, compared to directly fine-tuning on individual tasks, Routing Tuning achieves better or comparable results on the eight tasks, indicating that our Routing Tuning method has a positive effect on maintaining the generalization performance of the model.

### 4.2.3 Model Merging Performance

In the PEFT library, several adapter fusion methods are available, including CAT, Linear, SVD, TIES(Yadav et al., 2024), and Dare(Yu et al., 2024). TIES and Dare have different implementations using SVD or linear. CAT and linear concatenate or add LoRA adapters directly. SVD also decompses the LoRA matrices using SVD to further merge. Ties(Yadav et al., 2024) propose the TRIM, ELECT SIGN and MERGE procedure. It first randomly drops some parameters then using task vector sings to merge the paramters with the same signs to alleviate parameter collision. And the merging process will use SVD methods. DARE(Yu et al., 2024) drops some parameters at a drop rate p, then uses linear or SVD methods. We choose the remaining performance as the performance matrix, calculated by new score divided by original score. If the model refuses to generate anything or reaches the max_frame limit of 30 minutes of gameplay, the final score will be 0. We do not consider randomness in gym environments. We choose the default or recommand parameters for svd_clamp or density.

We choose four games to merge: ChopperCommand, Asterix, Battlezone, and CrazyClimber. The model concatenates or adds the adapters directly for CAT and Linear methods. The model generates nothing for all games in our scenario, so the final score is 0. The best performance is magnitude_prune on BattleZone, which achieves 0.203 remaining on the best game. However, our merge method achieved better performance retention, with a retention rate of 0.552 on the best-performing game.

Table 4: Overall performance of LoRA merging methods. Fine-tunning means mixing the dataset of 4 games and fine-tune one model together.

| Methods | GAME1 | GAME2 | GAME3 | GAME4 |
|---|---|---|---|---|
| CAT | 0 | 0 | 0 | 0 |
| Linear | 0 | 0 | 0 | 0 |
| SVD | 0.132 | 0 | 0 | 0 |
| Ties | 0.098 | 0 | 0 | 0 |
| Ties_svd | 0.086 | 0 | 0 | 0 |
| Dare_ties | 0 | 0 | 0 | 0 |
| Dare_linear | _0.173_ | 0 | 0 | 0 |
| Dare_ties_svd | 0.109 | 0.013 | 0 | _0.036_ |
| Dare_linear_svd | 0.069 | 0.022 | 0.05 | 0.013 |
| Magnitude_prune | 0.081 | _0.034_ | _0.203_ | 0 |
| Magnitude_prune_svd | 0 | 0 | 0 | 0 |
| Ours | **0.328** | **0.179** | **0.552** | **0.182** |
| Fien-tuning | 0.693 | 0.162 | 0.828 | 0.454 |

Table 5: Results of iterative merging different tasks LoRA. We first merge two games to get a new model. Then we add one game at a time iteratively.

| Number of Merged | GAME1 | GAME2 | GAME3 | GAME4 |
|---|---|---|---|---|
| 2 | 0.620 | 0.144 | N/A | N/A |
| 3 | 0.328 | 0.216 | 0.586 | N/A |
| 4 | 0.328 | 0.179 | 0.552 | 0.182 |

We then tested our iterative merging method by selecting five games for gradual merging: Asterix, ChopperCommand, BattleZone, CrazyClimber, and DemonAttack, where one game was merged at a time. The results are shown in Table 5. The newly formed model was subjected to multiple single-task evaluations after each merge. Compared to previous methods in the PEFT library, our merging approach demonstrated significant performance improvements. Our method alleviates the catastrophic degradation in single-task performance. Compared to the fusion methods provided in the PEFT library, our approach balances the performance on both new and previous tasks. Additionally, our outputs are less prone to disruption, avoiding empty or garbled content.

## 5 CONCLUSION

This paper presents a method for multitask learning using LoRA adapter fusion methods. In contrast to traditional model fusion approaches, we have developed a pipeline from fine-tuning to model merging. Through Routing Tuning, the model learns based on different losses for different data, reducing interference between the model's capabilities and new tasks. Subsequently, multiple LoRA adapters can be combined to represent modifications to the base model, and more importantly, new task adapters can be iteratively added. A general model capable of handling various tasks can be constructed by directly merging models and fused adapters. We run the experiments in Atari game scenarios because each game can be seen as a separate task, unlike general tasks. The experiment results have shown the remarkable performance of our methods and demonstrate the potential of using LoRA adapters for fusion to enable lifelong learning capabilities in models.

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

Table 6: The average norm of the activation values of the LoRA layers across different tasks.

|  | GAME1 | GAME2 | GAME3 | GAME4 | GAME5 | Average |
|---|---|---|---|---|---|---|
| Game Data | 0.187 | 0.191 | 0.19 | 0.179 | 0.185 | 0.186 |
| General Data | 0.137 | 0.149 | 0.152 | 0.131 | 0.154 | 0.142 |

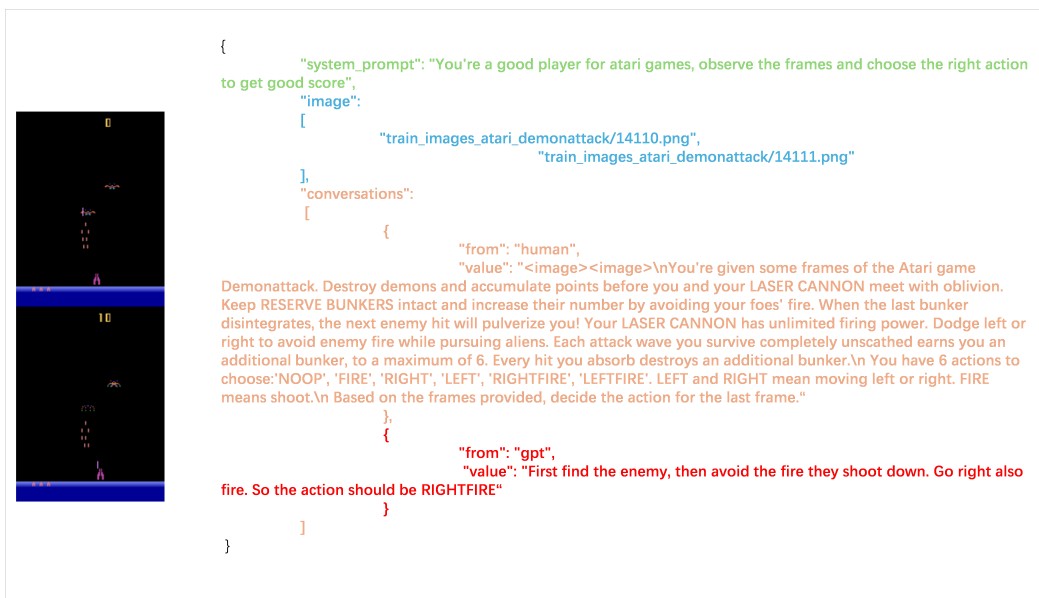

Figure 3: Visual and text input of VLM. Visual inputs are two frames sampled using APPO algorithm. And text inputs are the description of game rules and action meanings. Output will add a inference process, then we will use exactly match to get final action sent to gym environment.

## A   APPENDIX

### A.1   DATASET DETAILS

Our experiment uses game frames rendered from Atari environments as image input. To capture the character's movement direction and speed in the game, we selected two frames with $skip\_frame = 4$. The model needs to output the action to be taken in the latter frame. We extracted key game rules from the Atari official website's game descriptions, including the game background, character settings, rules, and gameplay. Additionally, since Atari games use the Atari hardware platform, mapping them to the OpenAI Gym environment involves predefined actions. Initial experiments showed that the model struggled to map action names to actual operations effectively. Therefore, we include specific actions in the prompt for each game. For output, we want the model to provide its reasoning about the game scenes and rules and then output an action. This approach achieved better results in our experiments. Details are shown in 3. We will make this dataset publicly available.

We sample data from LLaVA pretrianed data and Atari chanllenge data which is publicly available. For each game, we use 100K data we record using APPO checkpoint follow the baselines. For Atari challenge dataset, we have 4 games and each game we sample 5K image-action pairs. For LLaVA pretrained dataset, we sample 30K samples. Finally, we have 100K fine-tunning data and 50K general datato be unlearned.

### A.2   HYPERPARAMETER DETAILS

- Proportion of SVD rank: After testing on several games, we average the score to get the best performance. We find that, in some games, the portion saved of the SVD rank has few

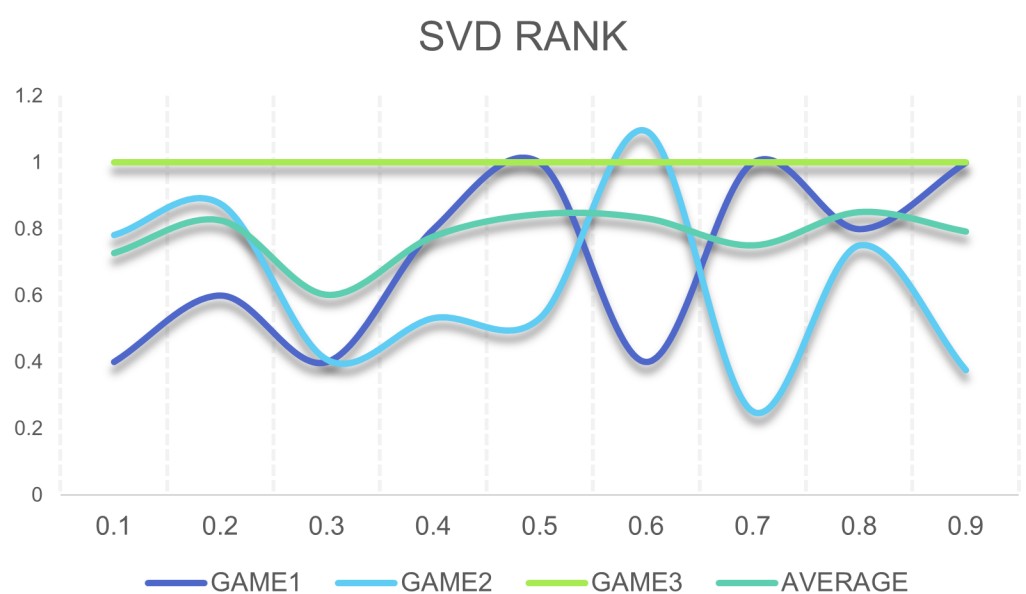

Figure 4: The proportion of rank retained by SVD across different tasks does not show a clear pattern of influence on the results. We selected 0.8, which yielded the highest average result.

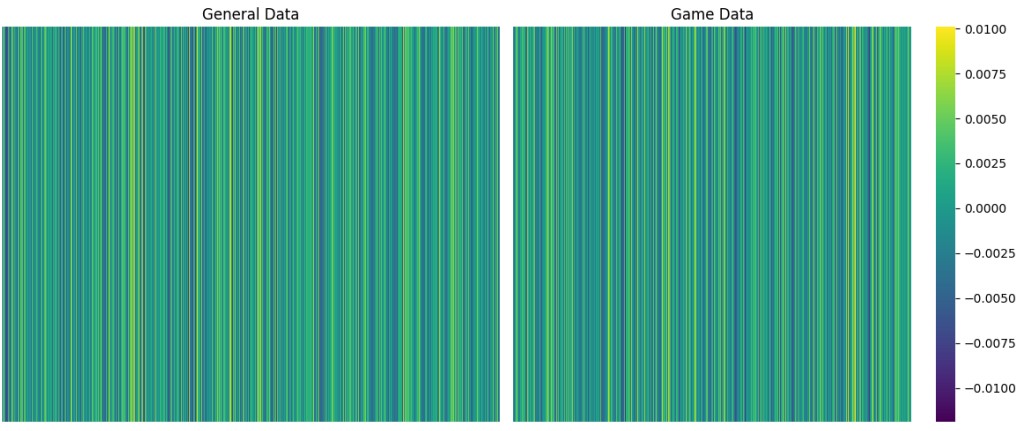

Figure 5: The heatmap of activation values for LoRA_B in the same layer across different data types shows that the proportion of higher values on general data is lower than that on game data. The specific numerical details are reflected in the table6.

    influence on the model performance. But we still choose the highest average score which is 0.8. The results are shown in 4

- $\epsilon$: We observe the output value of each component when losses are stable, then we choose to set the KL-divergence and norm on the same amount of fine-tuning loss, the final value is $\epsilon_1 = 1, \epsilon_2 = 0.01, \epsilon_3 = 2$

