# OpenReview forum: "I-Lora: Iterative Merging of Routing-Tuned Low-Rank Adapters for Multi-task Learning"
_ICLR.cc/2025/Conference — ICLR 2025 Conference Withdrawn Submission_

### Official Review · Reviewer_WxXV · 2024-11-02

**Soundness:** 2
**Presentation:** 3
**Contribution:** 2
**Rating:** 5
**Confidence:** 4

**Summary:**

This paper presents a method for fine-tuning VLMs in a continual learning setting, progressively adding new tasks via LoRA adapters. To preserve the original model's performance while fine-tuning for each new task, two additional losses are introduced: (i) a KL-divergence loss that aligns the logits of the fine-tuned model with those of the original model and (ii) a norm loss to push general task activations toward zero. When fine-tuning multiple tasks, the method uses SVD to iteratively merge the pairwise task-specific adapters into a single unified model.

**Strengths:**

The paper is clearly written, well-structured, and provides a strong motivation for the proposed method. Each stage of the process is well-explained, with experiments on pre-trained Atari tasks that demonstrate the approach’s effectiveness.

**Weaknesses:**

1. Comparison to Baselines in Experiments

While the proposed method is compelling, the choice of baselines and experimental setup could be improved for clarity. The Atari dataset is derived from a pre-trained APPO algorithm, more like treated as the expert dataset for distillation. Thus, the approach aligns more with multi-task distillation than traditional RL. Comparing the method directly to RL approaches like DreamerV3 and DART, which train from scratch, may not be fully appropriate. Instead, it would be more informative to present the original performance scores from the APPO algorithm and compare them to the distilled VLM's scores after each fine-tuning step. This would clarify how well the method preserves performance relative to the initial expert model.

2. Baseline Methods and Table 4 Clarifications

The explanation of the baselines in Table 4, especially the SVD method and the DARE method variants, is somewhat unclear. Providing a detailed description of these baselines and how they integrate into the benchmark would improve clarity, especially for readers who may be unfamiliar with each method. A brief explanation of each baseline’s structure, strengths, and limitations would also better highlight the unique challenges and contributions of the proposed approach.

3. Task Weight Balancing in Loss Function

The paper lacks discussion about task weighting (Equation 2) and how it impacts model performance. Specifically, how is the balance between general task data and fine-tuning data managed, and how sensitive is the method to this balance? A series of ablation studies exploring the effect of task weightings and data distribution would enhance understanding and demonstrate the robustness of the approach.

4. Direct Multi-Task Fine-Tuning Performance

It would be insightful to assess LoRA’s performance when applied directly to multi-task fine-tuning, as a reference point. This could help define an upper-bound performance level without SVD approximations, allowing readers to see the potential trade-offs introduced by the proposed merging approach and better appreciate the value of SVD in merging adapters without significant performance loss.

**Questions:**

See the weaknesses.

---

> ### Author Response · Authors · 2024-11-22
>
> Thank you for your detailed and thoughtful reviews. We greatly value your feedback, which has guided us in making meaningful improvements to the paper. We will make some clarifications for the weaknesses in the review.
>
> ---
> - Weakness 1
>     - The experimental results show that the performance of our VLM is weaker than the APPO method we used to collect data. However, it’s worth noting that the APPO method uses 2 billion frames of data per game, which is 20,000 times the amount of data we used. As such, this comparison might not be entirely fair. That said, there is still a significant gap between VLM trained on datasets without precise action labels and the state-of-the-art RL algorithms. On the bright side, our method does show advantages compared to the baselines used in the paper. Since there hasn’t been previous work reporting VLM performance on game data, our work serves as an exploration of VLM capabilities and provides a baseline for future research.
> - Weakness 2
>     - SVD decomposition and DARE include multiple implementation methods, and we tested all the implementations provided in the official PEFT library, using the official method names. It is true that we overlooked providing a detailed introduction to the baseline methods. We have now added this information in the section discussing the selection of experimental baselines.
> - Weakness 3
>     - In our experiments, we prioritized ensuring the effectiveness of the fine-tuning loss. Therefore, for the latter two parameters, we set their values based on the stabilized fine-tuning loss, ensuring consistency with it. We are currently conducting experiments on these parameters; however, due to limitations in computational resources, the results are not yet available. We will include this part of the experimental results in the final version of the paper to further demonstrate the effectiveness of our approach.
> - Weakness 4
>     - We have added the results of direct multi-task fine-tuning to Table 4 in our experiments. The results show that if the original training data is available and further pre-training or fine-tuning is possible, the performance of multi-task fine-tuning, while slightly reduced, is still better than directly merging the LoRA matrices. Our method is designed for scenarios where fine-tuned LoRA matrices are available, but the fine-tuning data itself is not accessible. In such cases, LoRA merging proves to be more practical and valuable.
>
>     |              | GAME1  | GAME2 | GAME3  | GAME4 |
>     |--------------|--------|--------|--------|--------|
>     | Our Method     | 0.328| 0.179| 0.552 | 0.182 |
>     | Fine-tuning| 0.693| 0.162| 0.828| 0.454 |
>
> ---
> We appreciate your understanding and hope the clarifications and revised manuscript meets your expectations.

---

### Official Review · Reviewer_QSW9 · 2024-11-03

**Soundness:** 1
**Presentation:** 2
**Contribution:** 2
**Rating:** 3
**Confidence:** 4

**Summary:**

This paper tackles the challenge of multi-task learning in vision-language models, specifically tailored for mastering multiple Atari games and general VQA tasks. It begins by gathering Atari data for training the vision-language model. Next, it introduces a method named Routing Tuning, which develops distinct LoRA adapters for various tasks. Finally, it presents an iterative maximum merging technique to consolidate these different LoRA adapters into a single one.

**Strengths:**

- Multi-task learning in vision-language models is a significant and intriguing area of study.

- The method is straightforward and easy to understand.

- Experiments demonstrate that multi-task learning using the proposed approach achieves performance comparable to single-task models, while maintaining a reasonable level of general vision-language understanding.

**Weaknesses:**

- In the experiments, is there an explanation for why the performance of some tasks improves after "routing tuning," while others decline compared to "single task fine-tuning" (see Table 2)?

- Similarly, in Table 3, the performance on most general tasks worsens after Routing Tuning, which seems to contradict the claim in Line 466 that "both single-task fine-tuning and Routing Tuning improve the model’s performance on general datasets." Am I misunderstanding this?

- The concept of "Maximize Merge" seems a bit unusual. I’m not saying it’s unfeasible, but what if we simply trained all game tasks together with general data in one model over more iterations? This important multi-task learning baseline is missing in Table 2.

**Questions:**

- Motivation: Why do we want to merge different LoRA adapters? Can we simply retain the task-specific adapter for this use case?


- I've noticed several typos, including those in lines 259, 261, and 433. The captions for images and tables could be more informative and helpful.

I am open to rating adjustment if my concerns discussed above are addressed.

---

> ### Author Response · Authors · 2024-11-22
>
> We sincerely thank the reviewers for their valuable feedback and insightful comments. Based on your suggestions, we have carefully revised and clarified the relevant sections to address the raised concerns.
>
> ---
> - Weakness 1
>     - In our paper, we mentioned that there is a random process in game testing. Some test rounds may result in very high scores, which can affect the overall score. We overcome this randomness by conducting multiple rounds of testing and averaging the scores. In our later model fusion, we also eliminated games with high randomness. Additionally, whether the model's unlearning could potentially enhance the learning of certain tasks remains a possibility, which we will investigate with more detailed testing.
> - Weakness 2
>     - Thank you for pointing out the error in Line 466. We acknowledge that it's a typo. This error does not affect the overall conclusions of our paper. The correct information should be [Both methods reduced the model's performance on general tasks, but our method resulted in significantly less reduction.]. This is corrected in the the paper.
> - Weakness 3
>     - If the number of parameters is limited, an infinite model fusion is certainly impossible. What we are exploring is how to integrate new tasks into previously trained tasks while mitigating forgetting, thereby investigating a possibility for continual learning.
> - Questions:
>     - This is the motivation for our paper. We hypothesize a scenario with numerous tasks that cannot be trained simultaneously. We propose leveraging their LoRA adapters for model merging.  Our approach offers advantages over methods like LoRAMoE, which retain multiple adapters for inference: 1. Our merged model ultimately retains the same parameter count as the original model, saving parameter storage space, GPU memory, and inference time. 2. We enable dynamic merging of new tasks and their adapters.  Maintaining multiple adapters and using them concurrently during inference requires retraining the weight-determining components when incorporating new tasks, introducing significant practical inconvenience.
>     - Thank you for pointing out the typos! We have corrected the relevant section and also enriched the content of the figure captions.

---

> > ### Comment · Reviewer_QSW9 · 2024-11-30
> > **Thank you for your response**
> >
> > I have reviewed the comments and responses from all reviewers, as well as the revised version of the submission. While I appreciate the authors’ efforts in addressing the typos and presentation issues I highlighted, the overall quality of the writing and the thoroughness of the experiments remain questionable. I believe there is significant room for improvement in the paper for future revisions.

---

### Official Review · Reviewer_jJq7 · 2024-11-03

**Soundness:** 2
**Presentation:** 1
**Contribution:** 1
**Rating:** 3
**Confidence:** 3

**Summary:**

This paper presents I-LoRA, a fine-tuning approach for vision-language models designed to overcome performance degradation associated with adapting models to new tasks. The authors highlight limitations in current vision-language models, which, despite improving visual and logical task capabilities, face challenges in continual learning due to high computational costs and reduced performance during fine-tuning.

I-LoRA addresses these issues by introducing:

1. A **Routing-Tuning** method that minimizes interference with the model's original capabilities when learning new tasks, keeping the activation of Low-Rank Adapter (LoRA) matrices low.
2. A **Merging Mechanism** for adapters, which supports continuous learning by allowing new tasks to be added without performance loss from task order interference.

Experiments on diverse datasets reportedly validate I-LoRA’s advantages, achieving strong single-task performance, effective multi-task compatibility, and scalability without parameter increases.

**Strengths:**

Empirical results show a considerable improvement in different games when compared to the selected baselines in Tables 1 and 3.

**Weaknesses:**

**Method**

- Routing: I am unsure about the novelty of maintaining the information learned from the vanilla VLM using a *data-driven* approach. It uses data similar to that used to train the based model. It requires training the LoRA with the target and the previous datasets to teach when intervening with the weights of the base model, which doesn't look ideal and is not generalizable since, for many VLM, we don't necessarily have access to the datasets that have been trained.

---

**Experiment**

- No ablation of the losses that are proposed? What are the $\epsilon_1$, $\epsilon_2$, and $\epsilon_3$, and what values do they take?


---

**Literature review**

A couple of papers on model merging are missing from the literature review, especially in L57-59 and L64-L66.

- [1] Yang, E., Wang, Z., Shen, L., Liu, S., Guo, G., Wang, X., & Tao, D. Adamerging: Adaptive model merging for multi-task learning, ICLR 2024

- [2] Zhang, F. Z., Albert, P., Rodriguez-Opazo, C., Hengel, A. V. D., & Abbasnejad, E.. Knowledge composition using task vectors with learned anisotropic scaling. NeurIPS 2024

- [3] Yang, E., Shen, L., Guo, G., Wang, X., Cao, X., Zhang, J., & Tao, D. (2024). Model merging in llms, mllms, and beyond: Methods, theories, applications and opportunities. arXiv preprint arXiv:2408.07666.

- [4] Ilharco, G., Ribeiro, M. T., Wortsman, M., Gururangan, S., Schmidt, L., Hajishirzi, H., & Farhadi, A. Editing models with task arithmetic. ICLR 2023

Also, concerning Adapters
.
- [5] Zhang, Q., Chen, M., Bukharin, A., He, P., Cheng, Y., Chen, W., & Zhao, T. Adaptive Budget Allocation for Parameter-Efficient Fine-Tuning. ICLR 2023

---

**Presentation**
- Figures and Tables have very poor captions.
- Typo: Llava L161, L347 and L430
- Table 1: you could add some colours to rows and distinguish when the method is better than human or SOTA.

---

**Questions:**

1. Equation 4. Is this making the activations approach zero? Do you have any visualization of it? Visualising the activations for a target and base datasets test set would be ideal.
2. L431. Why do you decide to use 30,000 samples from the general data? How do you sample the dataset?

---

> ### Author Response · Authors · 2024-11-22
>
> We sincerely appreciate your review and the valuable suggestions provided, which have significantly contributed to enhancing our paper. We offer a thorough explanation on your weakness and questions in our responses below. Thank you again for your efforts in helping us improve our work.
>
> ---
> - Weakness 1
>     - We do not require the use of the model's original training data. As long as the datasets to be unlearned are as general as possible and unrelated to the tasks to be learned, they are sufficient. The datasets we are adding here consist of the open-source datasets used by the LLaVA model and the open-source Atari datasets, and they do not include the training datasets used by the LLaVA-interleave model.
> - Weakness 2
>     - We first obtained the stabilized values for each corresponding part and ultimately selected a set of parameters that the loss of KL-divergence and Norm would not interfere with the loss during fine-tuning. That said, we agree that we should conduct a fine-tuning experiment on these parameters to ensure the completeness of the experimental results. This experiment is currently underway, and since the training process takes a considerable amount of time, we will include the results in the final version of the paper once they are ready.
> - Weakness 3
>     - I've read the relevant references you provided, and they are indeed related to our work. We have included them in the related work section and cite them.
> - Weakness 4
>     - Thank you for your questions regarding the paper’s representation. We have made the necessary adjustments including typos and captions of figures and tables.
>
> - Questions:
>     - Yes, in the appendix we included additional experiments for validation, selecting LLAVA training data and game data not present in our training set. We extracted the outputs of the LoRA layers and verified their average norms, finding that the norms for the games were significantly larger than those for LLAVA. We also created heatmaps to validate our hypothesis.
>
>     |              | GAME1  | GAME2  | GAME3  | GAME4  | GAME5  | Average |
>     |--------------|--------|--------|--------|--------|--------|---------|
>     | Game Data    | 0.187  | 0.191  | 0.19   | 0.179  | 0.185  | 0.186   |
>     | General Data | 0.137  | 0.149  | 0.152  | 0.131  | 0.154  | 0.142   |
>
>     - Thank you for your suggestion. Our methods is to keep the general data to half the amount of the task data, so we included 30,000 LLAVA data and 20,000 open-source Atari data. However, we haven't experimented with this data volume yet. We are conducting detailed experiments and will include the results in the updated version of the paper.

---

> > ### Comment · Reviewer_jJq7 · 2024-11-26
> >
> > Thank you for your response and agreeing that you should conduct ablation on the fine-tuning $\epsilon$ parameters to ensure the completeness of the experimental results. I strongly recommend comparing your method against Adamerging [1] and Atlas [2]. This comparison would contextualize your contributions and provide a clearer picture of your approach's strengths and limitations.
> >
> > I have seen the revised version of the paper. Thank you for pointing out the appendix. However, the rebuttal partially answers my concerns. The captions of the figures and table are barely improved. I would love to see a revised version with better adjustments for the presentation. An experiment section with the comparison and ablation is needed. The statement "We do not require the use of the model's original training data. As long as the datasets to be unlearned are as general as possible and unrelated to the tasks to be learned, they are sufficient" is a significant claim. It should be experimentally verified by an ablation study comparing results using the model's original training data versus a general dataset or at least showing how unrelated to the task it is. Such evidence is critical to support this assertion.
> >
> > Given these gaps, I believe the paper remains incomplete in its current state, and I must maintain my original score. I encourage the authors to address these points, as they are critical for establishing the robustness and generalizability of the proposed method.

---

### Official Review · Reviewer_MCvQ · 2024-11-03

**Soundness:** 2
**Presentation:** 3
**Contribution:** 2
**Rating:** 5
**Confidence:** 4

**Summary:**

This paper proposes an iterative LoRA merging method for multi-task learning. The authors employ Singular Value Decomposition (SVD) to reduce the number of redundant parameters and keep the components with the most significant influence to ensure task performance. The authors conduct experiments on Atari game and show that the proposed method outperforms state-of-the-art RL methods.

**Strengths:**

- The idea of iterative mering LoRA adapters is interesting.
- The proposed method achieves competitive results.

**Weaknesses:**

- The task formulation is not clear. According to the description in L67 'this paper focuses on enabling the model to learn new tasks while maintaining performance on other tasks.', the task is more likely to be continual learning/lifelong learning, rather than multi-task learning. However, in L158 the problem definition seems to be more like multi-task learning. The former focuses on learning a series of tasks and keeping the old task performance, i.e. alleviating catastrophic forgetting. Usually the old tasks are not accessible when learning new tasks. In contrast, the latter focuses on learning multiple tasks simultaneously. The authors should further clarify the problem definition in Section 3.1.
- The assumption of the proposed constraint is not convincing. The authors propose to constrain the LoRA's activation to zero when dealing with general or other tasks. What if the new task is beneficial for the general ability of the VLM? For instance, there are several studies investigating the forward transfer [1] in lifelong learning, and the task conflict in multi-task learning [3]. Therefore, the assumption behind is somewhat not convincing.
- The experimental setting is not convincing enough. Is there any specific reason to choose Atari Game? It seems that there is no public or widely-used Atari benchmark for VLM-based agent. Why not choose Minecraft, meta-world to verify the effectiveness of the proposed method? At least, there are lifelong learning agent baseline for Minecraft (VOYAGER) and multi-task RL baselines for meta-world [4,5].
- The experiment part is limited to VLA task, i.e. Atari Game. Does the proposed I-LoRA also apply to other general LLM/VLM Multi-task/Continual learning? It seems that the I-LoRA is not specifically designed for Game.
- Minor points:
    - More details of the general dataset should be provided.
    - The authors could adopt a more intuitive metric for Table 1, such as average ranking, or normalized average score.
    - The performance drop could be provided for clearer comparison in Table 3.
    - Baselines are mostly RL methods. The authors should compare with more baselines and variations, including vanilla VLM, PEFT-based methods, LoRA-merging methods and their variants.


[1] Gradient Episodic Memory for Continual Learning. NeurIPS 2017

[2] Beyond Not-Forgetting: Continual Learning with Backward Knowledge Transfer, NeurIPS 2022

[3] Gradient Surgery for Multi-Task Learning.	NeurIPS 2020

[4] Multi-task reinforcement learning with soft modularization, NeurIPS20

[5] Multi-Task Reinforcement Learning with Context-based Representations, ICML21

**Questions:**

- How is the ratio of kept singular value determined?
- What is the input of the VLM? The authors mentioned text input, how about the visual input?
- How are the game rules described?
- How is the COT conducted?
- How to make action to interact with the Atari environment from the VLM's output?

---

> ### Author Response · Authors · 2024-11-22
>
> We really appreciate the reviewer’s positive and constructive feedback. We will make clarifications to weaknesses and questions below, please let us know if there is anything we need to address further.
>
> ---
> - We appreciate the review from the reviewer. To make it clear, our focus is on continual learning by integrating each new task with previously learned models, ensuring minimal degradation in the model's performance on prior tasks while mastering the new one. Our experiments follow this approach: we fine-tune on a new game, merge the resulting new LoRA matrix into the previously integrated model, and then evaluate the model's performance on both the current and previous games. Ultimately, we achieve a model with the same amount of parameters to the original, capable of handling multiple games and retaining general capabilities well, while allowing for continuous integration of future tasks.
> - We assess the model's general performance by evaluating the benchmark results. If these benchmark results are reliable, our experiments indicate that fine-tuning on a large amount of single-game data does indeed lead to a decline in performance.
> We chose Atari because the tasks are clearly defined and significantly different from each other, making it suitable for testing multi-task fusioncapabilities. The reinforcement learning defined by VOYAGER requires using GPT to build a skill library, which makes it difficult to construct data for training VLMs for a single skill. Additionally, the required capabilities overlap between different skills. We will continue to identify suitable benchmarks for testing multi-task integration in VLMs.
> - On one hand, Atari game tasks are easier to divide into independent tasks, as the visuals and gameplay of each game differ significantly. This makes it an ideal setting to test the effectiveness of model merging. On the other hand, Atari still presents challenges for current VLMs, as the model needs to demonstrate capabilities such as multi-image comparison, combined text-image understanding, and reasoning. Our current fine-tuning results on some complex games are far below the human average score, even GPT-4o can’t perform well based on our early test, highlighting this as a promising direction for improving VLM capabilities. We have also noticed that some model merging approaches use more general tasks, and we plan to incorporate them into our future work.
> - Questions:
>     - SVD rank determined by our experiment on several single games, we choose the highest score remaining performance.
>
>     |       | 0.1       | 0.2   | 0.3       | 0.4       | 0.5      | 0.6      | 0.7   | 0.8   | 0.9       |
>     |-------|-----------|-------|-----------|-----------|----------|----------|-------|-------|-----------|
>     | GAME1 | 0.4       | 0.6   | 0.4       | 0.8       | 1        | 0.4      | 1     | 0.8   | 1         |
>     | GAME2 | 0.78125   | 0.875 | 0.40625   | 0.53125   | 0.53125  | 1.09375  | 0.25  | 0.75  | 0.375     |
>     | GAME3 | 1         | 1     | 1         | 1         | 1        | 1        | 1     | 1     | 1         |
>     | AVERAGE | 0.727083333 | 0.825 | 0.602083333 | 0.777083333 | 0.84375 | 0.83125 | 0.75  | 0.85  | 0.791666667 |
>
>     - The inputs contains two continuous frames of the game  and game rules
>     - We collect the rules from the official website, then use GPT for summarize, the final rules are determined by humans.
>     - Compared to directly output action token, we found that letting model output the description of the game state and action to take performs better. We give each action a meaning by playing the game with humans. So the COT output is：state description + action meaning + final action.
>     - By directly mapping because the action are all capital letters. (Details are shown in the appendix)

---

> > ### Comment · Reviewer_MCvQ · 2024-11-27
> > **Official Comment by Reviewer MCvQ**
> >
> > Thank you for your response. Part of my concerns are addressed. However, given the main goal of the paper, it would be better to reformulate the problem setting and include more commonly used continual-RL benchmarks such as Continual-world. I think the current version is not ready for publication, and I choose to keep my rating.

---

### Note · Authors · 2024-12-04

I have read and agree with the venue's withdrawal policy on behalf of myself and my co-authors.